# Citizen scientists: Unveiling motivations and characteristics influencing initial and sustained participation in an agricultural project

**Birgit Vanden Berghen** *, **Iris Vanermen**, **Liesbet Vranken**

Department of Earth and Environmental Sciences, Division of Bio-economics, KU Leuven, Leuven, Belgium

* birgit.vandenberghen@kuleuven.be

**Data Availability Statement:** The dataset used for this publication is stored at the Research Data Repository (RDR) by KU Leuven. They can be

## Abstract

Citizen science, where non-specialists collaborate with scientists, has surged in popularity. While it offers an innovative approach to research involvement, the domain of agri-environmental research participation, particularly in terms of citizen recruitment and retention, remains relatively unexplored. To investigate how what factors influence initial and sustained participation in an agronomic citizen science project, we performed a large survey during the case-study "Soy in 1000 Gardens". We obtained data on citizens motivations, general values, environmental concern, prior citizen science experience, and knowledge regarding sustainable food consumption and garden management and applied a two-step selection model to correct for potential self-selection bias on our participation outcomes. Initially, citizen scientists appear to be mostly motivated by gaining knowledge, having fun social interactions and environmental concern with regards to the effects on others, while the desire for enhancing or protecting their ego is less prominent. They also display higher knowledge and self-transcending values. Sustained participants however, are significantly older and share a stronger sense of moral obligation than their dropped-out counterparts. Moreover, prior experience seems to positively influence the length of their participation, while higher knowledge has a positive impact on the amount of data contributed. These insights offer strategies for tailored engagement that should emphasize collective impact, align with intrinsic values, and foster a sense of moral duty, with potential to enhance agri-environmental citizen science initiatives' effectiveness in addressing environmental challenges.

## 1. Introduction

Citizen science, a collaborative volunteering approach that engages non-specialists in scientific research, has witnessed a remarkable surge in popularity over the past two decades [1,2]. This participatory model of research fosters interactions between researchers and citizens, resulting in innovative changes in research methodologies and outcomes [3]. While citizen science encompasses a diverse range of research practices globally, it is particularly prominent in environmental sciences, including ecology and conservation [3,4–6]. On the other hand, a much

accessed via the following DOI: https://doi.org/10.48804/UYEYVY.

**Funding:** - L.V received funding for the research through the 'Soy in 1000 Garden Project'. B.V.B. was employed through this funding. The 'Soy in 1000 Gardens' project (GC03-C02) was funded by the Grand Challenges Program of VIB, which received support from the Flemish Government under the Management Agreement 2017-2021 (VR 2016 2312 Doc.1521/4). - https://vib.be/en - The sponsors played no role in the study design, data collection and analysis, decision to publish or preparation of the manuscript.

**Competing interests:** The authors have declared no competing interests.

smaller part of published citizen science research focusses on more applied fields like food or agriculture [7,8]. Agricultural citizen science projects often last a relatively long time period and require active involvement of the citizen scientist in different stages of the project. As such, the success of agricultural citizen science projects heavily relies on participation and sustained involvement by citizens.

While technological advancements have lowered the threshold for participation, recruitment and retention remain a challenge in citizen science [3,9–12]. Participation is complex in nature. There are multiple levels of participation that can depict a citizen's engagement in a citizen science project. In turn, these levels are often supported by different drivers [13–15]. By differentiating between participation levels when evaluating project engagement, it becomes possible to explore the distinct drivers underlying each level [15]. Collectively, research efforts have underscored the significance of a desire for learning and understanding, a commitment to conservation, and personal values for motivating participation in environmental citizen science projects [16–19]. Still, factors that drive people to start participating in a project, can diverge from factors that drive sustained participation in a citizen science project [20–22]. Project drop-out has been recorded for widely varying reasons on all levels of participation [13,15,23]. This is troubling since high participation and input from citizens is crucial for the outcome of these projects.

Hence, to ensure the success of citizen science projects, it is crucial to better understand what traits of citizens ensure participation throughout the whole project lifespan. This article aims to contribute to these insights in two ways. Firstly, we want to assess which motivations and personal traits are prevalent among people that initially participate in citizen science projects. Researchers may be able to more effectively target recruiting efforts if they have a better grasp of the traits of the people that engage in citizen science. Secondly, we want to know which of those personal traits and motivations are most relevant for sustained participation. In this way, the study seeks to inform recruitment and selection strategies, and foster long-term engagement in citizen science initiatives. The case study used for this research was the *Soja in 1000 Tuinen* (Soy in 1000 Gardens) project, a citizen science project involving over 1000 Flemish citizens in a six-month garden experiment focused on agricultural research and applications.

## 2. Theoretical background

### 2.1 Citizen science as a form of volunteering and pro-environmental behavior

Citizen science engages non-specialists in the process of creating new scientific knowledge, often in collaboration with professional scientists [1]. It entails citizens taking on diverse research roles, ranging from data collection to co-designing research processes while adhering to scientific standards. Similar practices are seen worldwide under terms like community science and community-based research, showcasing a broad spectrum from large-scale data collection to independent research efforts by non-academic groups [24]. In our case study, citizens primarily focused on data collection and generating scientifically valuable outcomes, a form of citizen science often termed as 'crowdsourcing' [24]. Volunteering literature has regularly been applied to citizen science. It is defined as planned, unpaid, pro-social behavior benefiting strangers. This aligns with citizen science where participants contribute time and effort without financial reward, often engaging in research led by distant scientists with minimal direct impact on participants [13,25].

Further, engagement in citizen science projects also highlights the significant overlap between agricultural citizen science and pro-environmental behavior, as these projects often

involve individuals actively engaging in practices that promote environmental sustainability [26,27]. By participating in agricultural citizen science initiatives, individuals not only contribute to scientific knowledge and innovation but also actively support sustainable farming practices, biodiversity conservation, and the promotion of resilient food systems [8]. This underscores the significant role that citizen science plays in fostering both scientific inquiry and pro-environmental behavior within the agricultural domain, and highlights the potential for citizen science to address contemporary agricultural challenges.

## 2.2 Quantifying participation in citizen science projects

Active engagement by volunteers is of crucial importance for the working success of any volunteer-based organization. In volunteering literature, Penner distinguishes between 'initial participation' and 'sustained participation' [25]. The former refers to what is often described as recruitment and allows us to assess the factors that set apart participants from non-participants. The latter terms refers to retention and the degree of participation which can be attributed to the volunteers [25]. Similarly to volunteering, the recruitment and retention of citizens is also of crucial importance for the success of a participatory research project. Within retention, literature has defined multiple levels of participation that can depict a citizen's engagement in a citizen science project [14,15,28]. For example, Eveleigh and colleagues were among the first to formulate distinct categories of citizen science participants, each attributed their own level of engagement and corresponding motivations, resulting in the three groups 'dropouts', 'dabblers' and 'super-users' [14]. Similarly, research on dropout rates of registered participants to *Biodiversity4all* categorized them based on their level of participation: 'never participating', 'occasional participation', 'regular participation', and 'frequent participation' [28]. Finally, a study by Fischer and his colleagues introduced the 'Nibble-and-Drop' framework. This framework, partly inspired by the work of Eveleigh, effectively described multiple dropout points and stages of contribution typical of participation in a citizen science project [15].

## 2.3 Motivations for participation in citizen science

Past research has explored the effect of motivations on initial and sustained citizen science participation [14,28–30]. Motivations are defined as the subjective reasons that individuals claim or recognize to be driving their behavior [30].

**Volunteer function inventory.** A widely used tool for understanding motivations is the Volunteer Functions Inventory (VFI). The VFI consists of six different volunteer functions representing reasons why people volunteer. These include: 'values'–a way to express ones altruistic and humanitarian values, 'understanding'–a way to gain knowledge, skills, and abilities, 'enhancement'–a way to help the ego grow and develop, 'career'–a way to improve career prospects, 'social'–a way to develop and strengthen social ties, and 'protective'–a way of protecting the ego from the difficulties of life [31].

Studies employing the VFI method to assess motivations for participation in citizen science projects have consistently highlighted the 'values' function as a predominant driver for initial participation in citizen science endeavors [13,18,29]. This indicates that people are intrinsically motivated to participate in citizen science projects because it allows them to express their altruistic values. However, projects with a large educational aspect tend to also engage people motivated by the 'understanding' and 'career' functions, as such eliciting a more egoistical aspect of initial participation [13,18,32,33]. Sustained participation in citizen science is often also positively associated with motivations to gain knowledge, skills and abilities (i.e. *understanding* in VFI) and or to express altruistic and humanitarian values (i.e. *values* in VFI), as demonstrated in several studies [18,34]. One even revealing a negative association with motivations to

improve career aspects or to develop and strengthen social ties (i.e. *social* and *career* in VFI) [34]. On the other hand, a study examining variations in VFI across different participation durations in a water quality monitoring program found that the 'career' function positively influenced sustained engagement for specific demographics, including nonwhite individuals, college students, lower-income earners, those employed in the environmental sector, and younger participants [35]. These findings emphasize the importance of aligning project objectives with participants' values and interests, while also recognizing the influence of socio-economic and cultural factors on sustained engagement.

**Environmental concern.**   Following the description of environmental volunteering motivations in the 2021 study by West and colleagues, environmental concern can also serve as a catalyst for increased participation in citizen science initiatives [33]. Theoretically, as individuals are more aware of environmental issues and the urgency of addressing them, when environmental protection aligns with their values, they are often motivated to take action [36]. Citizen science offers a tangible way for people to contribute to scientific research and sustainability efforts, empowering them to directly engage with environmental issues.

**Moral imperatives.**   A moral imperative is a strongly-felt principle that compels that person to act [37]. Several studies researchers have highlighted the nuanced ways in which moral imperatives motivate individuals to contribute their time and resources to volunteering and pro-environmental behavior [38–41]. This underscores the possible roles of moral obligation and a feeling of responsibility in shaping individuals' engagement in citizen science.

## 2.4 Dispositional variables influencing participation in citizen science

In the realm of volunteering literature, Penner also introduced the concept of "dispositional variables" [25]. While individuals may feel motivated to volunteer, their decision to engage in a particular project hinges on its compatibility with their lifestyle. According to Penner, this compatibility depends on "dispositional variables", which include personal attributes influencing an individuals' propensity to volunteer, such as personal values and beliefs, and knowledge, as well as demographic factors like age, income, and education [13,25]. As such, dispositional variables can influence both motivation to participate and participation itself.

**Socio-demographics and knowledge.**   When looking at the characteristics of volunteers, and more specifically citizen scientists, it is frequently found that they are male and higher educated, more affluent, more likely to live in rural areas, more likely to be middle-aged, and overwhelmingly white [13,42–44]. While knowledge itself might not be a direct predictor for participation, what knowledge is often correlated with, namely (higher) educational level and income, can be considered predictors [45,46]. Moreover, a theory called the knowledge deficit model suggests that individuals' lack of scientific knowledge or understanding about a particular topic can lead to a deficit in their ability to engage effectively with scientific issues [47,48]. This also implies that, when individuals learn about pressing scientific challenges or areas of interest, they may seek out opportunities to contribute to research efforts. This way, by aligning with their interests, objective scientific knowledge about specific topics can spark individuals' interest and motivation to participate in citizen science initiatives, potentially influencing both initial and sustained participation [49].

**Values.**   Values are the fundamental beliefs and principles that are deemed important in guiding your actions and choices [50]. A widely used measure of values is The Schwartz Theory of Basic Human Values, from hereon after referred to as 'general values', developed by S. Schwartz. The general values are a psychological framework that proposes a universal structure for understanding human values across different cultures. It consists of ten basic human values

that represent broad, enduring guiding principles of life organizing people's attitudes, emotions, and behaviors, and typically endure across time and situations [50].

Numerous studies have identified a positive correlation between three types of general values–'self-enhancing', 'self-transcending', and 'biospheric'–and pro-environmental behaviors [51–54]. Zooming in on participation in citizen science specifically, one study found that values associated with openness to change, such as self-direction, were significant factors in driving initial participation, while a variety of values, with the exception of power, contributed to sustaining involvement over time [55]. Moreover, the study suggested that values linked to ongoing participation are influenced by extrinsic motivations, implying that when external incentives align with self-directed goals, individuals are inclined to perform tasks eagerly and consistently.

**Prior experience.**   Previous volunteer experiences have been found to be positively associated with future volunteering [56,57]. This is consistent with the idea that individuals who have volunteered in the past are more likely to continue doing so. However, the quality of these experiences can vary, with some individuals reporting less satisfactory experiences, which can lead to increased stress and decreased commitment to the organization [58]. Therefore, while previous volunteer experiences may increase the likelihood of future volunteering, the quality of these experiences is also important.

## 3. Case-study: Soja in 1000 tuinen

Globally, production patterns linked to traditional meat and dairy production often place a heavy burden on the environment, and this has created a need for more plant-based protein sources. In North-Western Europe, in the last few years the focus has shifted to local and sustainable soybean production, with several citizen science initiatives helping the introduction of this high-protein crop [59–61]. Also Belgium, and more specifically Flanders, has followed this trend by setting up its own large scale citizen science project that focused on introducing soy as a successful crop in Flanders. Despite the breeding of numerous soybean varieties in Europe to optimize growth [62,63], a critical element essential for maintaining stable and high soybean yields is yet to be addressed—the bacterial inoculant. For reliable high-protein yields, soybean plants require effective interactions with nitrogen-fixing bacteria, commonly known as rhizobia, within their root nodules. Currently, commercially available rhizobia bacteria, sold as an inoculum, are not adapted to our (soil) climate and thus do not perform well enough.

In March 2021, The "Soja in 1000 Tuinen" project was launched [23]. To start 1,200 citizen were selected to plant soybean in their gardens. These citizens had two main tasks. Firstly, over the course of six months, from May to September, they reported on soybean growth and yield of ten soybean plants during five consecutive project steps. By registering data-input by the participants over the course of these steps, each participant's engagement was continuously measured (Fig 1). Secondly, halfway through the summer, participants returned five out of ten plants back to the lab, where professional scientists checked them for root nodules. The project led to a big phenotypical dataset for each garden that could directly be related to the occurrence of nodules, as well as a collection of bacteria isolated from these nodules. Ideally, several of these isolated strains will prove to nodulate soybean in both lab and field conditions, making them ideal candidates for a new and improved soybean inoculum for agriculture. Additionally, soil collected from each garden was analyzed to determine its nutrient, chemical and microbial composition. Combined with the phenotypical data, this provides a better fundamental insight into the environment needed for optimal soybean growth and nodulation. Lastly, the project aimed at increasing public awareness about the benefits of legumes and microbes for sustainable garden management and agriculture.

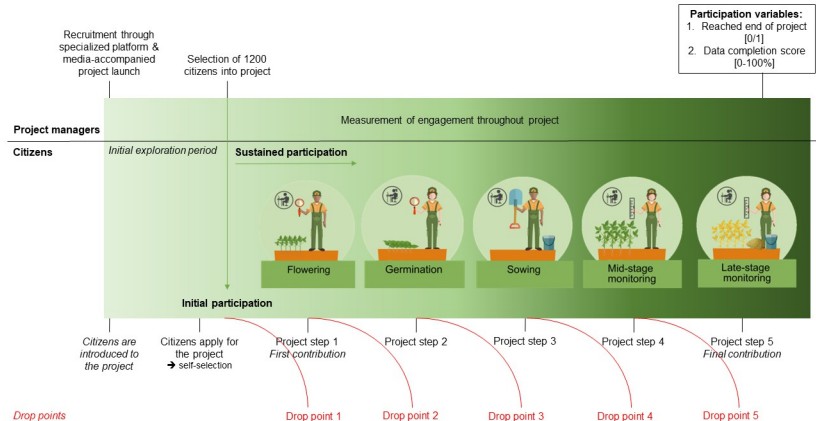

**Fig 1. Initial and sustained participation in the case study project offering insight from both the perspective of the project managers and the citizen scientists.** Visual representation of each of the five project steps within sustained participation. Red lines indicate potential drop-out between project steps. Partly based on the 'Nibble-and-drop' framework [15].

## 4. Material and methods

### 4.1 Participation

In Fig 1, the difference between initial and sustained participation within our citizen science project is visually represented, offering insight from both the perspective of the citizen scientists and the project managers. This depiction serves as a tool for understanding the stages described below.

**Initial participation–participant recruitment and selection.** Recruitment was carried out through two major pathways (Fig 1). Firstly, we collaborated with 'Mijn Tuinlab', an interactive platform for citizen science projects related to gardens, to recruit participants. This platform was chosen due to its community of a few thousand garden owners interested in citizen science projects. Secondly, the project launch included a press event with the Flemish Minister for Economy, Innovation, Work, Social Economy, and Agriculture in March 2021. The press event was featured in newspaper segments and on the national evening news, expanding the recruitment reach to the whole of Flanders. Registration was done via our project website, www.sojain1000tuinen.be, receiving a total of 5335 applications. Participants were informed about the project's objectives, including receiving information about their garden's composition and sustainable gardening practices, but no financial compensation.

Out of 5335 applicants, 1200 were representatively selected, each responsible for a garden plot in their own garden. The criteria for which we aimed at a representative sample included geographical distribution over Flanders and demographic factors, such as age, sex and educational level. Moreover, the use of fertilizer in participants' own gardens was only allowed in 20% of the participants, as fertilizer has a negative impact on the occurrence of root nodules. Participants had to be 18 years or older.

**Measurement of sustained participation.** To form a clear image of high or low contributions and long-term commitment within the project, for each participant sustained participation in the project was measured in two different ways (Fig 1).

1. 'Participation until the end' represents a binary variable which captures whether the participant participated till the end of the project. As such, it measures long-term commitment. For this variable, participants who reached the final step of the project, received a value of 1, while participants who did reach the final step received a value of zero. This meant that

participants who missed a step somewhere during the course of the project, due to vacation, forgetfulness, lack of time, etc. were still seen as a sustained participant as long as they were still actively submitting data in the final step.

2. 'Data completion score' represents the actual contribution made by the participant in the project by measuring the participants' average data completeness over five plants (or less if less than five plants were available due to biological reasons) over all five project steps. This variable ranged from 0% to 100%.

Defining sustained participation in two different ways, namely the length of their participation and the amount of input delivered during the project, allows us to look at different aspects of participation and determine whether the way participation is defined influences results.

## 4.2 Survey instrument

The citizens were recruited in the project from May 1$^{st}$ to October 31$^{st}$ 2021, and were asked to participate in an extensive survey at the start, between May 1$^{st}$ and May 14$^{th}$ 2021. The study protocol and survey were approved by and carried out in accordance with the guidelines and ethical standards outlined by the Social and Societal Ethics Committee (SMEC) of KU Leuven. Online, written informed consent was obtained from all participants included in the survey. Participants were provided with clear information about the purpose of the study, and their voluntary participation was emphasized. Confidentiality and anonymity of participants were strictly maintained throughout the survey process, and data were collected and reported in aggregate to ensure privacy. All data were handled and stored securely, and access was restricted to authorized personnel only. The survey design and execution adhered to the principles of fairness, respect, and integrity. All included motivations and dispositional variables were included based on a literature study, for reasons we touched upon in the introduction and theoretical background.

Fig 2 displays all factors included in the survey, classified following the example of West, assessing the influence of 'motivations' and 'dispositional variables' on citizen science volunteering [13,33].

**Motivations.** For motivations, we considered volunteering motivations from VFI [31], where environmental concern [64] could be considered an expression of the VFI factor 'values' [33], as well as moral imperatives including a feeling of responsibility for the current environmental decline [65] and a feeling of moral obligation to finish the project [66].

*Volunteer Function Inventory* Six variables from Clary's Volunteer Function Inventory were included in the questionnaire to measure individuals' motivations for engaging in volunteer activities [31]. The motives were measured as latent constructs, and consisted of five single items each, resulting in a thirty item segment. The items were translated from English to Dutch and included:

- Values–a way to express ones altruistic and humanitarian values.

- Understanding–a way to gain knowledge, skills, and abilities.

- Enhancement–a way to help the ego grow and develop.

- Career–a way to improve career prospects.

- Social–a way to develop and strengthen social ties.

- Protective–a way of protecting the ego from the difficulties of life.

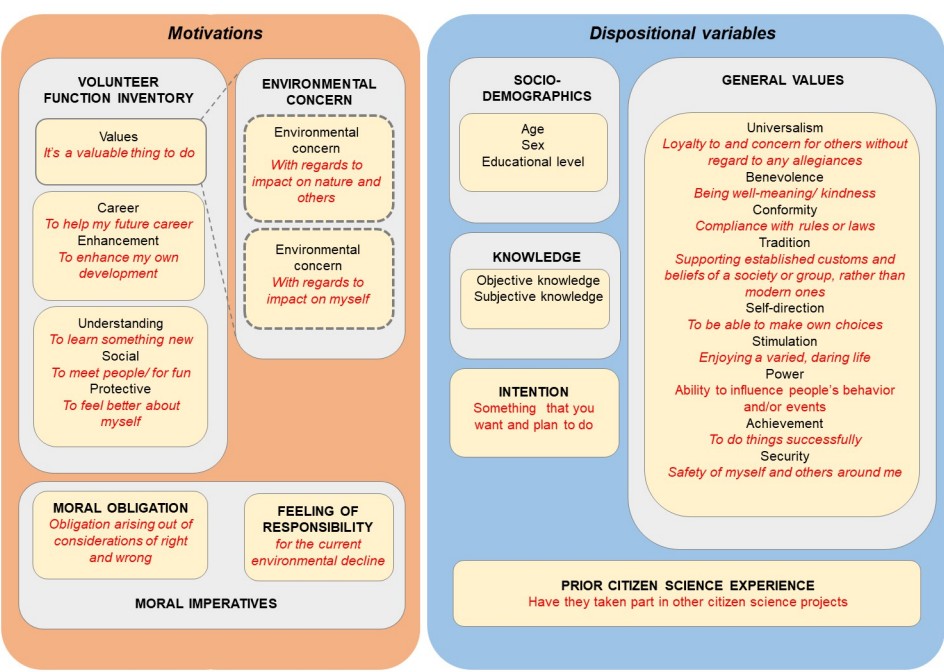

**Fig 2. Classification of all influencing 'motivations' and 'dispositional variables' following the example of West (2016, 2021).** Specifically, volunteering motivations from VFI, together with environmental concern, and moral imperatives form 'Motivations' for participating in the citizen science case study 'Soja in 1000 Tuinen'. Socio-demographics, knowledge, general values, participation intention and prior citizen science experience make up 'Dispositional variables' which also possibly influence participation in the project, but are distinct from motivations. All variables were included based on a literature study, for reasons touched upon in the theoretical background.

The respondents were asked to indicate on a scale from 1 ("Totally not agree") to 7 ("Totally agree") to what extent they agreed with the item statements. Each latent construct consisted of the average value of its five respective items.

*Environmental concern* Three variables from Schultz's 'Environmental Concerns' were included in the questionnaire [63]. The variables were measured as latent constructs, and consisted of three single items each, resulting in a nine-item segment, and included: 'Concern for myself', 'Concern for others', and 'Concern for the environment'. The respondents were asked to indicate on a scale from 1 ("Not at all") to 7 ("Extremely") how concerned they were about the listed items. The items were translated from English to Dutch. Each latent construct consisted of the average value of its three respective items.

*Moral imperatives* One variable was included to measure the feeling of personal responsibility for the current environmental decline in the survey. Participants answered the question "How responsible do you feel about the current environmental problems on a scale from one to ten?". This item was derived from the Norm-Activation Model [64]. Secondly, another variable consisting of three items on moral obligation with regard to volunteering were included in the questionnaire (e.g. "I would feel guilty if I didn't volunteer for this citizen science project."). The items were answered on a 7-point scale from "Totally don't agree" to "Totally agree". They were based on the moral extension of Theory of Planned Behavior, adapted for volunteering in citizen science projects and translated from English to Dutch [65]. The latent construct 'moral obligation' consisted of the average value of its three respective items.

**Dispositional variables.** Our dispositional variables consisted of socio-demographics, knowledge, general values [50], intention to complete the project, and lastly prior citizen science experience.

*Socio-demographics* Questions about gender, age and education level were included in the questionnaire.

- Gender was questioned and translated to a binary variable, with a value of 1 if the participant was female, and a value of 0 if the participant was male.

- Education level was measured through the highest obtained degree at the moment of participation, and divided into four categories: primary education, secondary education, bachelor, and master or more. A new binary variable was then created, indicating whether participants had obtained a degree in higher education (1) or not (0).

  Knowledge

- Objective knowledge: One variable consisting of twenty items was included to measure objective knowledge on sustainability [67]. Each item was a short multiple choice question. Ten items were related to sustainable food consumption, and ten items were related to sustainable garden management. Per item, participants could obtain a 0 (wrong answer) or 1 (correct answer) value. The measure for objective knowledge was obtained by adding the score for all 20 items, and dividing this value by 2, resulting in an 'average out of 10' score.

- Subjective knowledge: One variable on subjective knowledge was included consisting of two items measuring subjective knowledge of (1) sustainable food and (2) sustainable garden management, respectively: "How much do you know about to food and sustainability on a scale from one to ten?", and "How much do you know about to gardening and sustainability on a scale from one to ten?". The items were answered on a scale from 1 to 10, and adapted from [67] for sustainable food consumption and sustainable garden management. The measure for subjective knowledge consisted of the average value of the two items.

*General values* Nine variables to measures individuals' personal values across different dimensions were included in the questionnaire to provide insight into the importance that individuals place on these different values. The variables were measured as latent constructs, and consisted of two to four single items each, resulting in a thirty-seven item segment, and included: 'Universalism', 'Power', 'Self-direction', 'Tradition', 'Benevolence', 'Achievement', 'Stimulation', 'Conformity', and 'Security'.

- Universalism–Loyalty to and concern for others without regard to any allegiances.

- Power–Ability to influence people's behavior and/or events.

- Self-direction–To be able to make own choices.

- Tradition–Supporting established customs and beliefs of a society or group, rather than modern ones.

- Benevolence–Being well-meaning/ kindness.

- Achievement–To do things successfully.

- Stimulation–Enjoying a varied, daring life.

- Conformity–Compliance with rules or laws.

- Security–Safety of myself and others around me.

The respondents were asked to indicate on a scale from 1 ("Not important at all") to 7 ("Extremely important") how important the items were as values in their everyday life. The

items were translated to Dutch from the adapted Schwartz Theory of Basic Human Values [51,68]. Each latent construct consisted of the average value of their respective items.

*Intention* One variable consisting of three items on intention with regard to volunteering were included in the questionnaire (e.g. "I am committed to volunteering in this citizen science project that aims to introduce soy as a sustainable source of protein."). Items were answered on a 7-point scale from "Totally not agree" to "Totally agree." The items were based on the Theory of Planned Behavior, adapted for volunteering in citizen science projects and translated to Dutch [69]. The latent construct consisted of the average value of its three respective items.

*Prior experience* One variable questioning whether or not participants had any prior experience with participating in a citizen science project was included. The variable was answered with 'Yes' or 'No', and afterwards translated to a binary variable 1 or 0, respectively.

### 4.3 Data collection

Participation and data input related to the soybean plants was continuously monitored between the start and end of the project in in spring/summer 2021. Additionally, at the start of the project, cross-sectional data was collected through an extensive survey. The survey was administered online to a sample of 1700 adults in Flanders over the age of 18. Of these, 1200 respondents were participants in the 'Soja in 1000 Tuinen'-project. Additionally, a control group, including 500 citizens who were not a part of the project, was recruited through an external market research agency. They were recruited from the Flemish population, with similar criteria as the project participants, namely a representative sample regarding age, sex and educational level, with participants older than 18 years. The survey data was collected using an in-house survey software of one of the project partners, Vlaams Instituut voor Biotechnologie (VIB), Forms for the citizen scientists and Qualtrics software for the general public. Survey responses were downloaded into Stata 17 for analysis.

As the complete questionnaire was considered lengthy, a deletion procedure was applied to remove respondents that showed fatigue and disinterest. This was done by removing respondents that had no dispersion in their answers on entire questions or filled in less than 90% of the answers. Additionally, participants whose reason for dropping out was "biological reasons resulting in plant death" were dropped from the analysis as those people had to stop their participation in the project involuntarily. The remaining sample consisted of 1161 respondents of which 809 were participants in the project, and 352 were member of the general public. Table 1 renders the sample description, displaying spread regarding gender, average age and educational attainment of both the citizen scientists and general public control group.

### 4.4 Data analysis

Through Confirmatory Factor Analysis (CFA) we tested whether measures of the latent constructs were consistent with our understanding of the nature of that construct. As such, CFA

**Table 1. Descriptive statistics of the socio-demographic variables.** Gender and educational attainment are expressed in absolute numbers.

| | | Citizen scientists | General public |
|---|---|---|---|
| Gender | Male | 442 | 189 |
| | Female | 367 | 163 |
| Age | Mean | 52.2 ± 14.5 | 49.0 ± 16.0 |
| Educational attainment | Primary | 4 | 7 |
| | Secondary | 240 | 168 |
| | Bachelor | 264 | 99 |
| | Master + | 301 | 78 |

was used as a method to verify the fit of the proposed latent factors and evaluate the reliability and validity of these factors. Due to the sensitivity of the chi-square test to sample size, additional fit indices were examined. The Comparative Fit Index (CFI), the Tucker-Lewis Index (TLI), the Root Mean Square Error of Approximation (RMSEA) and the Standardized Root Mean Square Residual (SRMR). Given an initial modest model fit for some factors, a few modifications were made by adding correlated error terms. To assess internal consistency, Alpha Cronbach scores of each latent factor were calculated. Table 2 represents the index coefficients indicating the model fit.

To get a preliminary notion of which motivations and dispositional variables differ significantly between (1) initial participants (citizen scientists) and the general public and (2) sustained and dropped-out participants, independent samples t-tests and subsequent Cohen's d tests were performed in Stata (Version 17.0). Cohen's d is a measure of effect size that compares the means of two groups. While a p-value can tell whether or not there is a statistically significant difference between two groups, an effect size can tell how large this difference actually is. Cohen's is interpreted as follows: Small effect: Cohen's d around 0.2, Medium effect: Cohen's d around 0.5, Large effect: Cohen's d around 0.8 or higher [70].

A Heckman selection model was then used to investigate which factors predicted initial participation in citizen science projects, as well as sustained participation. This model allows to address potential biases stemming from self-selection in a case study project. As touched upon in the theoretical background, environmental citizen scientists display specific motivations and personal traits [13,34,37,43]. By voluntarily applying to join a citizen science project, participants effectively self-select themselves into the pool of citizen scientists, potentially leading to biases in estimation results of the outcome variables [71,72].

The Heckman model employs a two-equation system to correct for this bias—one focusing on selection into the sample (opt-in when an outcome is observed–the sample selection equation), and the main equation linking the covariates of interest to the outcome (the outcome equation) [73]. The first equation, the selection equation, employs a logit model to estimate the likelihood of inclusion in the citizen scientist sample based on observable characteristics. The second equation, the outcome equation, examines the relationship between variables of interest and sustained participation within the selected sample. By integrating variables related to opting-in and sustained participation, a potential correction is done for sample selection bias from self-selection into the case study project, enabling more robust and reliable estimation of parameters of interest [74,75]. The outcome equation is specified as follows:

$$y_i^* = \beta x_i + \varepsilon_i$$

$$y_i = 1 \; if \; y_i^* > 0, \; y_i = 0 \; otherwise \tag{1}$$

$$y_i = y_i^* \; if \; y_i^* > 0, \; y_i = 0 \; otherwise \tag{2}$$

**Table 2. Minimum index coefficients of goodness of fit indicators specifying acceptable or great model fit.**

|  | Acceptable fit | Great fit |
|---|---|---|
| CFI | > .85 | > .95 |
| TLI | > .85 | > .95 |
| RMSEA | < .08 | < .05 |
| SRMR | < .08 | < .05 |
| Alpha cronbach | < .60 | < .90 |

where $y_i^*$ = is an unobserved latent variable that determines the likelihood of a citizen scientist participating in the project completing all steps or filling in all required data, and depends on a vector of motivations $x_i$ and random error $\varepsilon_i$. Actual sustained participation $y_i$ can take two forms based on the variables discussed in section 4.4:

- (1) Participation until the end: $y_i$ is either positive (1) or zero (0), depending on whether $y_i^*$ is above zero or not. This means that the traditional Heckman Two Stage Sample Selection Model for continuous variables cannot be used. As such, a Two Step probit model will be used to measure potential sample selection bias.

- (2) Data completion score: $y_i$ is either positive ($y_i^*$) or zero (0), depending on whether $y_i^*$ is above zero or not. As data completeness is a continuous variable, a traditional Heckman Two Stage Sample Selection Model will be used.

Within the outcome model we included Age, Intention, Moral obligation, Universalism, Benevolence, Power, Achievement, Self-direction, Stimulation, Tradition, Conformity, Security, Environmental concern with regards to impact on nature, Environmental concern with regards to impact on myself, Environmental concern with regards to impact on others, Feeling of responsibility, Objective knowledge, and Subjective knowledge as variables.

The dependent variable for the sample selection equation must be a binary variable, as the decision being modelled is an individual's choice to participate in a citizen science project or not. The sample selection equation is specified using a probit regression and estimated using maximum likelihood as follows:

$$s_i^* = \alpha x_i + \gamma z_i + \mu_i$$

$$s_i = 1 \; if \; s_i^* > 0, \; s_i = 0 \; otherwise$$

where $s_i^*$ is an unobserved latent variable that determines the likelihood of project participation for individual i, and depends on a vector of observed characteristics $x_i$, a vector of exclusion restrictions $z_i$, and random error $\mu_i$.

Within the sample selection model we included Age, Values, Understanding, Enhancement, Career, Social, Protective, Universalism, Benevolence, Power, Achievement, Self-direction, Stimulation, Tradition, Conformity, Security, Environmental concern with regards to impact on nature, Environmental concern with regards to impact on myself, Environmental concern with regards to impact on others, Feeling of responsibility, Objective knowledge, and Subjective knowledge as variables.

To ensure non-collinearity between the outcome equation and the selection equation, the selection equation must include an observed variable, $z_i$ that affects why individuals may select to participate in study but does not influence the outcome variable. This variable is referred to as the exclusion restriction. Here, the exclusion restriction is the subjective knowledge variable. Individuals who rate their own knowledge on project-related topics highly may feel more confident and motivated to participate in a citizen science project at the outset. While their perception of expertise or interest in the subject matter may drive them to engage with the project initially, based on literature, insignificant t-tests, and subsequent regression analysis we hypothesize that this has no impact on sustained participation.

In Heckman selection models, the traditional R-squared measure commonly used in ordinary least squares (OLS) regression models isn't directly applicable because the Heckman model involves two stages. However, one can still assess model fit or goodness of fit in Heckman models using alternative methods. One common approach is to use the Likelihood Ratio (LR) test of independent equations (rho = 0) [76]. This is a statistical test that assesses the joint

significance of the selection equation and the outcome equation being independent. In other words, it tests whether there is evidence to reject the null hypothesis that there is no correlation between the errors in these equations (i.e., rho = 0). The LR test is based on comparing the log-likelihood of the full model (allowing for correlation between errors) to the log-likelihood of a restricted model (omitted selection equation, thus assuming no correlation between errors). A significant LR test suggests that the full model with correlation between errors provides a better fit to the data than the restricted model [73,76].

## 5. Results

### 5.1 Confirmatory factor analysis

Table 3 summarizes the results of the goodness of fit analyses for a one-factor model per latent variable. The final goodness of fit indices indicated a very good model fit for all factors. Reliability analysis revealed mostly high internal consistency for all latent factors with Cronbach's alpha coefficients all measuring above 0.6. Moreover, all items loaded higher than 0.4 on their factors with most of them loading above 0.6 (S1–S3 Tables). This indicates that all items can be considered practically significant. Therefore, all items were considered in the interpretation of the factors.

### 5.2 Differences between citizen scientists and the general public

T-tests and corresponding Cohen's d values revealed statistically significant differences in almost all variables between the citizen scientists and the general public (Table 4). Firstly, citizen scientists were significantly older and often scored higher on variables associated with

**Table 3. Goodness of fit indices (CFI, TLI, SRMR, RMSEA) and Alpha cronbach for motivations and dispositional variables which were measured using a Likert-scale.**

|     |                         | CFI   | TLI   | SRMR  | RMSEA | Alpha cronbach |
| --- | ----------------------- | ----- | ----- | ----- | ----- | -------------- |
| VFI | Values                  | 0.960 | 0.900 | 0.040 | 0.129 | 0.7965         |
|     | Understanding           | 0.973 | 0.933 | 0.034 | 0.096 | 0.7673         |
|     | Enhancement             | 0.949 | 0.974 | 0.027 | 0.099 | 0.8359         |
|     | Career                  | 0.997 | 0.994 | 0.012 | 0.042 | 0.8977         |
|     | Social                  | 0.982 | 0.954 | 0.027 | 0.092 | 0.8309         |
|     | Protective              | 0.987 | 0.975 | 0.020 | 0.082 | 0.8801         |
| EC  | Nature                  | 0.998 | 0.994 | 0.007 | 0.062 | 0.9388         |
|     | Myself                  | 1.000 | 1.000 | 0.008 | 0.000 | 0.8636         |
|     | Others                  | 1.000 | 1.000 | 0.027 | 0.000 | 0.8750         |
| MI  | Moral obligation        | 1.000 | 1.000 | 0.000 | 0.000 | 0.6465         |
| GV  | Universalism            | 0.996 | 0.977 | 0.015 | 0.055 | 0.6878         |
|     | Benevolence             | 1.000 | 1.000 | 0.004 | 0.000 | 0.7666         |
|     | Power                   | 0.996 | 0.989 | 0.014 | 0.037 | 0.7073         |
|     | Achievement             | 0.996 | 0.988 | 0.013 | 0.043 | 0.7361         |
|     | Self-direction          | 1.000 | 1.000 | 0.002 | 0.000 | 0.6606         |
|     | Stimulation             | 0.981 | 0.953 | 0.022 | 0.076 | 0.7664         |
|     | Tradition               | 1.000 | 1.000 | 0.010 | 0.000 | 0.5667         |
|     | Conformity              | 0.994 | 0.981 | 0.016 | 0.051 | 0.7341         |
|     | Security                | 0.983 | 0.948 | 0.025 | 0.082 | 0.7249         |
|     | Participation intention | 1.000 | 1.000 | 0.000 | 0.000 | 0.8051         |

VFI = Volunteer Function's Inventory, EC = Environmental concern, MI = Moral imperatives, GV = General values.

**Table 4. Results of t-test between citizen scientist (sample size: 809) and general public (sample size: 352) mean for all motivations and dispositional variables.**

| | | Citizen scientists | | General public | | | | | |
| --- | --- | --- | --- | --- | --- | --- | --- | --- | --- |
| | *Variable* | *Mean* | *std. dev* | *Mean* | *std. dev* | *t-test* | *p-value* | *Cohen's d* | *[90% conf. interval]* |
| VFI | Values | **5.33** | **(0.92)** | 5.01 | (1.10) | -4.99 | 0.00 | -0.32 | -0.42–0.21 |
| | Understanding | **5.01** | **(1.00)** | 4.90 | (1.14) | -1.62 | 0.05 | -0.10 | -0.21 0.02 |
| | Enhancement | 4.38 | (1.18) | **4.73** | **(1.22)** | 4.58 | 0.00 | 0.29 | 0.19 0.42 |
| | Career | 2.91 | (1.41) | **3.80** | **(1.44)** | 9.81 | 0.00 | 0.63 | 0.52 0.73 |
| | Social | 3.62 | (1.19) | **3.81** | **(1.25)** | 2.41 | 0.01 | 0.15 | 0.05 0.26 |
| | Protective | 2.81 | (1.30) | **3.82** | **(1.41)** | 11.83 | 0.00 | 0.76 | 0.65 0.86 |
| EC | Nature | **6.05** | **(0.93)** | 5.27 | (1.14) | -12.25 | 0.00 | -0.78 | -0.89–0.67 |
| | Myself | 4.94 | (1.17) | 5.01 | (1.11) | 0.96 | 0.34 | 0.06 | -0.04 0.17 |
| | Others | **5.79** | **(1.00)** | 5.22 | (1.11) | -8.53 | 0.00 | -0.54 | -0.65–0.44 |
| MI | Feeling of responsibility | **6.04** | **(1.80)** | 5.68 | (1.97) | -3.09 | 0.00 | -0.20 | -0.30–0.092 |
| SD | Age | **52.24** | **(16.04)** | 49.00 | (14.47) | -3.39 | 0.00 | -0.22 | -0.32–0.11 |
| KN | Objective | **8.80** | **(2.91)** | 6.51 | (2.83) | -12.47 | 0.00 | -0.79 | -0.90–0.68 |
| | Subjective | 9.96 | (3.56) | **10.45** | **(3.32)** | 2.15 | 0.02 | 0.14 | 0.03 0.24 |
| GV | Universalism | **5.68** | **(0.69)** | 5.00 | (0.95) | -13.63 | 0.00 | -0.87 | -0.98–0.76 |
| | Benevolence | **5.81** | **(0.70)** | 5.50 | (0.87) | -6.41 | 0.00 | -0.41 | -0.52–0.30 |
| | Power | 3.43 | (1.00) | **3.91** | **(0.99)** | 7.56 | 0.00 | 0.48 | 0.37 0.59 |
| | Achievement | 4.50 | (0.94) | 4.53 | (0.94) | 0.60 | 0.54 | 0.04 | -0.07 0.14 |
| | Self-direction | **5.53** | **(0.72)** | 5.10 | (0.88) | -8.87 | 0.00 | -0.57 | -0.67–0.46 |
| | Stimulation | **5.14** | **(0.82)** | 4.95 | (0.89) | -3.45 | 0.00 | -0.22 | -0.33–0.11 |
| | Tradition | **5.13** | **(0.76)** | 5.05 | (0.80) | -1.57 | 0.06 | -0.10 | -0.21 0.05 |
| | Conformity | 5.14 | (0.90) | **5.25** | **(0.92)** | 2.03 | 0.03 | 0.12 | 0.02 0.23 |
| | Security | 4.91 | (1.00) | **5.23** | **(0.92)** | 5.06 | 0.00 | 0.32 | 0.22 0.43 |

Bold variable results indicate a significantly higher answer for the group belonging to the bold variable. Effect size was calculated using Cohen's d. VFI = Volunteer Function's Inventory, EC = Environmental concern, SD = Socio-demographics, MI = Moral imperatives, KN = Knowledge, GV = General values.

increasing general welfare. For example, with regards to motivations, this included the VFI construct 'values' (i.e. it is a valuable thing to do) and understanding (i.e. to learn something new). Further, citizen scientists seemed to have a higher concern for the current environmental decline and how this could impact both nature, as well as people they care about and future generations (others). This concern was also accompanied by a higher feeling of responsibility for the current environmental decline than the general population. With regards to dispositional variables, for the general values constructs this included 'universalism' (i.e., loyal and concerned for others without regard to any allegiances) and 'benevolence' (i.e. being well-meaning and kind). Citizen scientists also scored significantly higher on the purely intrinsic general values related to individual welfare such as 'self-direction' (i.e., being able to make own choice) and 'stimulation' (i.e. enjoying a varied and daring life'). Finally, citizen scientists scored significantly higher on objective knowledge, but no significant difference was found for subjective knowledge between both groups.

The general public on the other hand often scored higher on VFI constructs related to egoism and self-importance, including 'enhancement' (i.e., enhance own development), 'career' (i.e., help future career), 'social' (i.e. to meet people/for fun), and 'protective' (i.e., to feel better about myself). These variables are all categorized as extrinsic motivators, although 'social' and 'protective' could also come from an intrinsic standpoint. The general public also scored higher on the general values functions 'power' (i.e., ability to influence someone's behavior

and/or events), 'conformity' (i.e., compliance with rules and laws), and 'security' (i.e., safety of myself and others around me).

## 5.3 Differences between sustained and dropped-out participants

Based on the 'End' variable, t-tests revealed statistically significant differences sustained and dropped-out participants in some variables (Table 5). Citizen scientists who participated throughout the whole project up until the end are significantly older than citizen scientists who dropped out at some point during the project and never made it to the final step. With regards to motivations, sustained participants scored significantly lower on the 'career' construct of the VFI. No other significant differences in VFI related motivations were found, however, sustained participants did show a higher moral obligation to finish the project. Finally, they also scored significantly higher on concern regarding the environment and other people with regards to climate change than citizen scientists who dropped out. With regards to dispositional variables, for the general values constructs sustained participants scored higher on 'universalism' (i.e. loyal to and concerned for others without regard to any allegiances) as well as 'conformity' (i.e. compliance with rules or laws) and 'security' (i.e. safety of myself and others around me). Lastly. sustained participants had a significantly higher objective knowledge, but no significant differences were found for subjective knowledge between both groups.

**Table 5. Results of t-test between sustained (sample size: 421) and dropped out (sample size: 388) citizen scientists for all motivations and dispositional variables.**

|  | | Participation until end | | Drop-out | | | | | |
|---|---|---|---|---|---|---|---|---|---|
|  | Variable | Mean | (std. dev) | Mean | (std. dev) | t-test | p-value | Cohen's d | [90% conf. interval] |
| VFI | Values | 5.31 | (0.93) | 5.34 | (0.90) | -0.38 | 0.70 | 0.03 | -0.09 0.14 |
|  | Understanding | 5.01 | (1.00) | 5.02 | (1.01) | 0.25 | 0.80 | 0.02 | -0.10 0.13 |
|  | Enhancement | 4.36 | (1.21) | 4.41 | (1.15) | 0.63 | 0.53 | 0.04 | -0.07 0.16 |
|  | Career | 2.81 | (1.39) | **3.01** | **(1.43)** | 2.05 | 0.04 | 0.14 | 0.03 0.26 |
|  | Social | 3.60 | (1.23) | 3.64 | (1.16) | 0.43 | 0.67 | 0.03 | -0.09 0.15 |
|  | Protective | 2.78 | (1.33) | 2.84 | (1.27) | 0.73 | 0.46 | 0.05 | -0.06 0.17 |
| EC | Nature | **6.11** | **(0.90)** | 5.98 | (0.95) | -1.91 | 0.03 | -0.13 | -0.25 0.02 |
|  | Myself | 4.90 | (1.17) | 4.99 | (1.18) | 1.03 | 0.31 | 0.07 | -0.04 0.19 |
|  | Others | **5.86** | **(0.93)** | 5.71 | (1.08) | -2.20 | 0.01 | -0.16 | -0.27–0.04 |
| MI | Feeling of responsibility | 5.98 | (1.78) | 6.12 | (1.82) | 1.09 | 0.28 | 0.08 | -0.04 0.19 |
|  | Moral obligation | **4.29** | **(1.23)** | 3.94 | (1.15) | -4.10 | 0.00 | -0.29 | -0.40–0.17 |
| SD | Age | **55.37** | **(14.24)** | 48.84 | (13.95) | -6.58 | 0.00 | -0.46 | -0.58–0.35 |
| KN | Objective | **9.00** | **(2.84)** | 8.58 | (2.98) | -2.04 | 0.02 | -0.14 | -0.26–0.03 |
|  | Subjective | 9.99 | (3.46) | 9.93 | (3.67) | -0.22 | 0.83 | -0.02 | -0.13 0.10 |
| GV | Universalism | **5.73** | **(0.67)** | 5.62 | (0.69) | -2.35 | 0.00 | -0.17 | -0.28–0.05 |
|  | Benevolence | 5.80 | (0.71) | 5.82 | (0.70) | 0.25 | 0.80 | 0.02 | -0.10 0.13 |
|  | Power | 3.44 | (1.00) | 3.42 | (1.00) | -0.34 | 0.73 | -0.02 | -0.14 0.09 |
|  | Achievement | 4.50 | (0.94) | 4.50 | (0.94) | -0.05 | 0.96 | 0.00 | -0.12 0.11 |
|  | Self-direction | 5.53 | (0.70) | 5.54 | (0.75) | 0.11 | 0.91 | 0.00 | -0.11 0.12 |
|  | Stimulation | 5.10 | (0.80) | 5.17 | (0.83) | 1.02 | 0.31 | 0.07 | -0.04 0.19 |
|  | Tradition | 5.15 | (0.77) | 5.10 | (0.75) | -0.89 | 0.37 | -0.17 | -0.28–0.05 |
|  | Conformity | **5.19** | **(0.86)** | 5.09 | (0.94) | -1.58 | 0.06 | -0.11 | -0.23 0.00 |
|  | Security | **4.98** | **(0.96)** | 4.84 | (1.03) | -1.91 | 0.03 | -0.13 | -0.25 0.02 |
|  | Participation intention | **4.62** | **(0.58)** | 4.52 | (0.62) | -2.53 | 0.01 | -0.18 | -0.30–0.06 |

Bold variables indicate a significantly higher answer for the group belonging to the bold variable. Effect size was calculated using Cohen's d. VFI = Volunteer Function Inventory, EC = Environmental concern, MI = Moral imperatives, SD = Socio-demographics, KN = Knowledge, GV = General values, V = Volunteering.

The same t-tests were conducted for investigating disparities between high- and low-contributing participants based on the data completion score. For this, each citizen was assigned a dummy score of 0 or 1 indicating whether their 'Data completion score' was situated below (low contribution) or above (high-contribution) the project mean, respectively. Results were practically identical to the t-tests based on the 'End' variable (S4 Table).

## 5.4 Two-step models

Table 6 displays the results of two Two-Step models examining the relationship between sustained participation and several variables, while correcting for the self-selected sample of citizen scientists.

**Participation until the end.** Within motivational VFI functions, 'understanding' and 'social' had a significant positive effect on becoming a citizen scientist, while 'enhancement' and 'protective' had a significant negative effect. This means that the higher citizens scored on being driven by gaining knowledge ('understanding') and meeting new people ('social'), and the lower they scored on being driven by enhancing ('enhancement') or protecting their ego ('protective'), the bigger the chance of them being/becoming a citizen scientist. Moreover, citizen scientists also scored significantly higher on environmental concern with regard to concern about the effects on other people, and significantly lower on environmental concern with regard to concern about the effects on themselves. With regard to dispositional variables, for general values, 'universalism' and 'achievement' both had a positive significant effect on being a citizen scientist, while 'security' had a negative effect. This indicates that citizen scientists generally scored higher for values and norms associated with 'universalism' such as broad-mindedness, protecting the environment and unity with nature, and 'achievement' such as success, ambition, influence and capability. On the other hand, they attached lower value to values and norms associated with 'security' such as social order and national and family security. Finally, objective knowledge on sustainable food consumption and garden management had a significant positive effect on becoming a citizen scientist while subjective knowledge had a significant negative effect.

With regards to the outcome equation, the model shows both age and moral obligation as having a strong significant positive effect on reaching the end of the project. Moreover, prior experience in citizen science also seemed to have a positive effect on whether or not a participant reached the end of the project.

In the overall model output, the Wald chi-square statistic ($\chi^2$) tests the overall significance of the independent variables in a model. Here, $\chi^2(18) = 77.44$ with a p-value of 0.0000, indicating that the participation till end model as a whole is statistically significant. Further, the coefficient value for rho is -0.600, with a p-value of 0.022. Rho indicates the correlation between the degree in which unobserved variables affect the selection equation, and the degree in which the same unobserved variables affect the outcome equation, expressed as the correlation between their error terms. A negative coefficient suggests a negative correlation, indicating an inverse relationship between the effect of unobserved variables on the selection and outcome. Further, the coefficient for rho statistically significant at a significance level of 0.1 (p-value = 0.022). This implies that there is reason to suspect that the relationship between factors affecting sample selection and the factors affecting the outcome variable is systematic. In other words, sample selection bias might be present and a two-step selection model is appropriate.

**Data completion score.** The data completion model showed identical results to the first model with regards to significant variables in the selection equation. Further, for the outcome equation, age and moral obligation also showed a strong, positive effect on the outcome variable. However, unlike the first model, objective knowledge seemed to have a significant,

**Table 6. Results of the Two-Step Selection Models.**

| | Participation until end | | | | Data completion | | | |
| --- | --- | --- | --- | --- | --- | --- | --- | --- |
| | Selection model | | Outcome model | | Selection model | | Outcome model | |
| | Coeff. | (std. err.) | Coeff. | (std. err.) | Coeff. | (std. err.) | Coeff. | (std. err.) |
| VFI—Values | -0.014 | (0.080) | -0.054 | (0.095) | -0.020 | (0.080) | -2.912 | (1.885) |
| VFI—Understanding | 0.212*** | (0.076) | -0.033 | (0.074) | 0.216*** | (0.076) | 0.071 | (1.787) |
| VFI—Enhancement | -0.187** | (0.073) | -0.014 | (0.069) | -0.183** | (0.073) | 0.544 | (1.565) |
| VFI—Career | -0.072 | (0.053) | 0.007 | (0.061) | -0.072 | (0.053) | 0.234 | (1.274) |
| VFI—Social | 0.100* | (0.055) | -0.061 | (0.049) | 0.102* | (0.055) | -1.473 | (1.285) |
| VFI—Protective | -0.232*** | (0.053) | 0.004 | (0.055) | -0.233*** | (0.053) | 0.839 | (1.448) |
| EC—Nature | 0.079 | (0.064) | 0.032 | (0.064) | 0.078 | (0.064) | 0.933 | (1.633) |
| EC—Myself | -0.122** | (0.060) | -0.101 | (0.052) | -0.117** | (0.060) | -2.110 | (1.302) |
| EC—Others | 0.156** | (0.065) | 0.107 | (0.065) | 0.149** | (0.065) | 1.025 | (1.626) |
| MI—Feeling of responsibility | -0.002 | (0.028) | -0.044 | (0.027) | -0.003 | (0.028) | -0.959 | (0.674) |
| MI- Moral obligation | | | 0.157*** | (0.045) | | | 3.802*** | (1.128) |
| SD—Age | 0.001 | (0.004) | 0.015*** | (0.004) | 0.001 | (0.004) | 0.449*** | (0.094) |
| SD—Gender | -0.169 | (0.104) | -0.068 | (0.096) | -0.170 | (0.104) | 0.533 | (2.461) |
| SD—Higher education | 0.050 | (0.105) | 0.012 | (0.103) | 0.037 | (0.106) | -1.077 | (2.655) |
| KN—Objective | 0.187*** | (0.022) | 0.027 | (0.017) | 0.187*** | (0.022) | 0.920** | (0.449) |
| KN—Subjective | -0.137*** | (0.018) | | | -0.136*** | (0.018) | | |
| GV—Universalism | 0.691*** | (0.093) | -0.026 | (0.098) | 0.701*** | (0.093) | -0.033 | (2.601) |
| GV—Benevolence | -0.013 | (0.114) | -0.209 | (0.104) | -0.005 | (0.115) | -1.766 | (2.638) |
| GV—Power | -0.150* | (0.078) | 0.069 | (0.070) | -0.148* | (0.077) | 1.351 | (1.814) |
| GV—Achievement | 0.274*** | (0.087) | -0.031 | (0.078) | 0.274*** | (0.087) | -0.254 | (2.022) |
| GV—Self-direction | 0.074 | (0.100) | 0.013 | (0.091) | 0.065 | (0.101) | 1.590 | (2.343) |
| GV—Stimulation | -0.075 | (0.094) | -0.023 | (0.082) | -0.069 | (0.091) | -3.151 | (2.108) |
| GV—Tradition | 0.153 | (0.104) | -0.042 | (0.087) | 0.165 | (0.104) | -2.821 | (2.229) |
| GV—Conformity | -0.083 | (0.104) | 0.113 | (0.088) | -0.080 | (0.105) | 2.900 | (2.250) |
| GV—Security | -0.494*** | (0.091) | 0.063 | (0.075) | -0.500*** | (0.091) | 2.075 | (1.987) |
| Participation intention | | | 0.117 | (0.084) | | | 1.479 | (2.115) |
| Prior experience | | | 0.174* | (0.098) | | | 3.902 | (2.439) |
| Constant | -2.064*** | (0.500) | -1.129 | (0.693) | -2.084*** | (0.505) | 40.065** | (18.375) |
| Rho | -0.600** (0.194) | | | | -0.423* (0.200) | | | |
| Wald χ²(18) | 77.44 | | | | 89.79 | | | |
| Prob > χ² | 0.000 | | | | 0.000 | | | |
| LR test of Indep. Eqns. | 5.55 | | | | 3.74 | | | |
| Prob > χ² | 0.019** | | | | 0.053* | | | |
| Observations | 1.147 | | | | 1.147 | | | |

*** p < 0.01.

** p < 0.05.

* < 0.1.

Standard error in parentheses.

VFI = Volunteer Function's Inventory, EC = Environmental Concern, MI = Moral imperatives, SD = Socio-demographics, KN = Knowledge, GV = General Values.

positive influence on data completion. This implies that people who were more knowledgeable, had a higher chance of providing data during the project.

In the overall model output, the Wald chi-square test statistic $\chi^2(18) = 89.79$, with $p = 0.000$, indicating a significant relationship between the independent variables and data

completion. Further, the estimated value for rho is -0.423, with p-value = 0.064. Similar to the first model, this negative value suggests a negative correlation between the error terms in the selection equation and the outcome equation, implying an inverse relationship between the effect of unobserved variables on the selection and outcome the significance of rho at a 10% level. This implies that there is a systematic relationship between the factors affecting selection into the citizen science pool and the factors affecting how much required data participants provide for the project. In other words, there is possible sample selection bias and the usage of a Heckman selection model to correct for this is appropriate.

**Estimation of correction.**   The first selection model, using 'end' as an outcome variable, yielded a chi-square value of 5.55 with a corresponding p-value of 0.019. The second selection model, using 'data completion' as an outcome variable, yielded a chi-square value of 3.74 with a corresponding p-value of 0.053. Both outcomes suggest that the assumption of independence between the selection and outcome equations can be rejected at a significance level of 10%. This implies that for this level the full two-step model is appropriate and provides a significantly better fit to the data than the restricted model without selection equation.

## 6. Discussion

Citizen science engages non-specialists in collaborative scientific endeavors, facilitated by advances in technology. While its popularity has surged, challenges persist in recruiting and, even more so, retaining participants. Projects vary widely, encompassing astronomy, environmental sciences such as ecology and conservation, and less commonly, fields like agriculture. The 'Soy in 1000 Gardens' project adds to a growing body of knowledge on how to effectively engage citizens to contribute to agricultural research with a societal impact through insights on both initial and sustained participation.

By using Two-Step models, we could explore the relationship between sustained participation and various motivations, correcting for any bias that came from citizens self-selecting into the pool of applicant prior to selection by project managers. Previous research has already shown how similar fields are affected by this potential self-selection bias. For example, in environmental valuation surveys two-step models correct for higher environmental values and/or concerns of self-selected survey respondents [76,77]. Two-step methods have also previously been applied in both volunteering and citizen science specifically, given that the decision to volunteer is made by individuals, indicating those who choose to volunteer constitute a self-selected sample rather than a random sample [71,72]. In our study, the significance of rho and the Likelihood Ratio tests validating the suitability of using two-step models. Our findings also reinforce the importance of employing two-step models in citizen science research regarding motivations and other factors influencing participation if possible, to enhance the accuracy of analyses pertaining on participation.

From the sample selection equation of the two-step models some assumptions can be made on a profile for initial participants in agri-environmental citizen science. Similarly to prior research, our study illustrates how citizens taking part seemed to be motivated by the opportunity to learn something new, as well as socialize people and have fun [14,78–80]. Moreover, our results also confirmed the positive influence of environmental concern with regards to effect on others in motivating initial participation [37,81]. On the other hand, initial participants were specifically not motivated by enhancing their own development, feeling better about themselves through participation, or by environmental concern with regards to the effect on themselves. While some studies underscored that citizens tend to mostly engage when projects offer personal value or benefit, one could argue that our results indicate agricultural citizen scientist to be mostly driven by intrinsic rather than extrinsic motivations [33,82].

With regards to dispositional factors indirectly influencing initial participation, we found that agricultural citizen scientists attach high value to loyalty to and concern for the welfare of others, while also finding it important to have achievements in life. This differs from the findings of [55], where they found openness-to-change related factors, such as 'self-direction' and 'stimulation' to be most important. Still, high self-transcending values explain why environmental citizen scientists may be more inclined to participate in projects that contribute to the greater good and tackle environmental concerns. Citizen scientists also demonstrated higher objective knowledge, indicating a better understanding of the subject matter having a positive influence on being selected into the sample of agricultural citizen scientist, thereby seemingly backing the assumption of the deficit model [47].

While participants can display high motivation at the project's start, maintaining their engagement over an extended period of time poses a challenge to almost any citizen science project [13,83]. From the outcome equations of the two models, we can make some assumptions about important characteristics and drivers for sustained participation in our case study. Both age and moral obligation consistently emerged as a strong significant positive predictor of sustained participation. This indicates that older participants were more likely to be engaged throughout the project, which is in line with previous findings [83,84]. Moreover, one could argue this is intuitively logical, since people who are older, especially older-middle aged, generally have more free time and thus more time to spend on the project. Moral obligation to participate in the project displayed a strong positive effect on sustained participation, seemingly confirming that a sense of moral duty may be a powerful driver for participants to stay committed [38–41]. Additionally, the outcome variable 'Data completion' was positively influenced by objective knowledge, while 'Participation until end' was positively influenced by prior experience. Interestingly, this could imply that the citizens who have participated in citizen science projects before are more likely to see it through until the end, while citizens who specifically contributed more of the required data, have a deeper understanding of environmental challenges and the project's objectives and thus may be driven by this.

Comparing the results of the selection equations with the outcome equations, we find that the motivations and dispositional variables that determine selection into citizen science are not the same factors that ensure continued participation in the project. Even more surprisingly, is that our study found no discernible influence from VFI motivations on sustained participation. This is interesting, since often a priori selection of participants is recommended based on motivations, and intrinsic motivations normally result in more sustainable participation compared to extrinsic motivations [20,79]. This discrepancy in our results challenges the assumption that motivations alone drive sustained engagement. Similarly, [34] discovered that motivations tied to backing scientific research diminished in significance as participants continued their engagement. Still, citizen science contributors are likely to remain engaged and committed if they perceive their motivations for contributing are being met by the project [18,19]. Our results could suggest a potential mismatch between project objectives and participant expectations.

Once individuals overcome structural or technical barriers to engage in a project, constraints like limited time availability and decreasing interest in the topic often become primary reasons for dropout [13,44]. This provides interesting opportunities for follow-up research regarding volunteer retainment, such as additional motivations or intervention treatments during these types of project. In accordance, recent research has shown continued involvement could requires extrinsic factors such as feedback, a sense of belonging, etc. For this, a range of design interventions have been proposed to prevent drop-out in citizen science projects, including emails addressing motivational factors, personalized feedback realized through

gamification elements such as badges and leaderboards, creation of an online community, etc. [85–88].

## Limitations and implications

Previous studies suggest that citizen scientists are often late to middle -aged men [9,49]. While in our study, 'age' and 'gender' did not seem to have a significant effect on the initial participation, this could be due to participant selection criteria set by researchers to ensure a gender, age, educational and geographically balanced participating sample. Imposing selection criteria, such as striving for a balanced sample encompassing diverse age groups, genders, and educational levels, on citizen scientists may present both limitations and advantages. On one hand, such criteria might restrict the pool of participants, potentially excluding individuals who could contribute valuable insights but don't meet the specified demographic requirements. However, on the other hand, enforcing these criteria can counteract biases inherent in citizen science projects, which often tend to attract higher-educated, older males. By actively seeking representation from a broader demographic spectrum, these criteria promote inclusivity, enhance the robustness of data analysis, and ensure that findings are more reflective of society as a whole. Thus, while imposing such selection criteria may pose challenges, it ultimately fosters a more comprehensive and equitable approach to citizen science.

With regards to recruitment or initial participations, questions could also be asked about who from the "general population" was not reached, and who was reached but did not participate. Since the project launch was accompanied by a large media campaign involving the national evening news, and national and regional newspaper, one could argue that project managers did their best to reach as large and diverse of a crowd as possible. However, it has been proven that current citizen science projects often do not come close to engaging a broad and diverse public. Though demographic variables did not have a significant influence on initial participation due to the selection criteria we set, no other demographic data such as ethnicity, income, disability, etc., were questioned. This has limited our knowledge on the possible lack of engagement from specific demographic groups, which could be very relevant recruitment during future projects. Studies such as [42], who identify potential targets groups of participants who could become more engaged, seem promising to fill in this gap, yet it remains unclear whether future CS projects have the potential to reach a wider range of participants.

## 7. Conclusion

Our results have implications for the design and implementation of future citizen science initiatives. Firstly, the significant influence of self-selection bias into the citizen science sample should be considered, especially upon interpretation of the influence of motivations and dispositional variables when a Heckman procedure is not possible due to the lack of a general public control group. Secondly, understanding the motivations and characteristics of citizen scientists can help project organizers tailor their recruitment strategies and engagement techniques. Emphasizing the collective and environmental impact of citizen science projects and aligning with participants' intrinsic values may enhance initial participation. Finally, efforts to foster a sense of moral duty could further increase participants' retention to the project.

Overall, citizen science initiatives hold great potential for addressing environmental challenges and advancing scientific research. The study's findings provide valuable insights into the differences between initial and sustained participation in citizen science projects. Further research could explore the long-term impacts of sustained citizen participation and investigate additional factors influencing engagement in diverse citizen science contexts.

## Supporting information

**S1 Table. Factor analysis showing R-squared and factor loading for latent variables of Volunteer Factor Inventory (VFI) and intention and moral obligation to participate in the project.**
(DOCX)

**S2 Table. Factor analysis showing R-squared and factor loading for latent variables of General Values (GV).**
(DOCX)

**S3 Table. Factor analysis showing R-squared and factor loading for latent variables of Environmental Concern (EC).**
(DOCX)

**S4 Table. Results of t-test between above average (sample size: 444) and below average (sample size: 365) contributors for all motivations and dispositional variables.** Bold variables indicate a significantly higher answer for the group belonging to the bold variable. Effect size was calculated using Cohen's d. VFI = Volunteer Function Inventory, EC = Environmental Concern, MI = Moral imperatives, SD = Socio-demographics, KN = Knowledge, GV = General Values.
(DOCX)

## Acknowledgments

The authors thank all volunteers, including those candidates who were not admitted to the project, for their enthusiasm to take part in the citizen science project and the collection of the survey and garden data. We also thank all partners and co-researchers on the project: VIB-UGent Center for Plant Systems Biology (Prof. Sofie Goormachtig, Prof. Steven Maere, Dr. Stien Mertens, Dr. Lena Vlaminck, Dr. Sonia Garcia Mendez, Alexander Clarysse), ILVO (Prof. Isabel Roldán-Ruiz, Dr. Joke Pannecoucque, Margo Vermeersch, Koen Van Loo), Ghent University (Prof. Anne Willems, Helena Van den Eynde, Ilse de Baenst, Prof. Caroline De Tender), VIB-KU Leuven (Prof. Jan Michiels, Serge Beullens), VIB HQ (Dr. Astrid Gadeyne, Dr. Laurien Van Dyck, Merlijn Vanhecke, Teknorix), MijnTuinlab, Bodemkundige Dienst van België.

## Author Contributions

**Conceptualization:** Birgit Vanden Berghen, Iris Vanermen, Liesbet Vranken.

**Data curation:** Birgit Vanden Berghen.

**Formal analysis:** Birgit Vanden Berghen.

**Investigation:** Birgit Vanden Berghen.

**Methodology:** Birgit Vanden Berghen, Iris Vanermen, Liesbet Vranken.

**Project administration:** Birgit Vanden Berghen, Liesbet Vranken.

**Resources:** Liesbet Vranken.

**Software:** Birgit Vanden Berghen.

**Supervision:** Iris Vanermen, Liesbet Vranken.

**Visualization:** Birgit Vanden Berghen.

**Writing – original draft:** Birgit Vanden Berghen.

**Writing – review & editing:** Iris Vanermen, Liesbet Vranken.

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
