## [Decision Letter · Decision Letter 0]

13 Feb 2024

PONE-D-23-34026What defines a citizen scientist and influences their sustained participation in an agronomic project?PLOS ONE

Dear Dr. Vanden Berghen,

Thank you for submitting your manuscript to PLOS ONE. After careful consideration, we feel that it has merit but does not fully meet PLOS ONE’s publication criteria as it currently stands. Therefore, we invite you to submit a revised version of the manuscript that addresses the points raised during the review process.

This is an interesting paper which deals with an important issue: namely the determinants sustained of citizen engagement in citizen science activities in relation to agronomy. However, reviewer 1 has s have indicated that there issues in relation to methodological clarity which need to be addressed, recommending major revision. Reviewer 2 has also raised some minor concerns to be addressed, 

Please submit your revised manuscript by Mar 29 2024 11:59PM. If you will need more time than this to complete your revisions, please reply to this message or contact the journal office at plosone@plos.org. Please include the following items when submitting your revised manuscript:A rebuttal letter that responds to each point raised by the academic editor and reviewer(s). You should upload this letter as a separate file labeled 'Response to Reviewers'.A marked-up copy of your manuscript that highlights changes made to the original version. You should upload this as a separate file labeled 'Revised Manuscript with Track Changes'.An unmarked version of your revised paper without tracked changes. You should upload this as a separate file labeled 'Manuscript'.

We look forward to receiving your revised manuscript.

Kind regards,

Lynn Jayne Frewer, MSc PhD

Academic Editor

PLOS ONE

[The authors thank all volunteers, including those candidates who were not admitted to the project, for their enthusiasm to take part in the citizen science project and the collection of the survey and garden data. We also thank all partners and co-researchers on the project: VIB-UGent Center for Plant Systems Biology (Prof. Sofie Goormachtig, Prof. Steven Maere, Dr. Stien Mertens, Dr. Lena Vlaminck, Dr. Sonia Garcia Mendez, Alexander Clarysse), ILVO (Prof. Isabel Roldán-Ruiz, Dr. Joke Pannecoucque, Margo Vermeersch, Koen Van Loo), Ghent University (Prof. Anne Willems, Helena Van den Eynde, Ilse de Baenst, Prof. Caroline De Tender), VIB-KU Leuven (Prof. Jan Michiels, Serge Beullens), VIB HQ (Dr. Astrid Gadeyne, Dr. Laurien Van Dyck, Merlijn Vanhecke, Teknorix), MijnTuinlab, Bodemkundige Dienst van België. The ‘Soy in 1000 Gardens’ project (GC03-C02) has received funding from the Grand Challenges Program of VIB, which received support from the Flemish Government under the Management Agreement 2017-2021 (VR 2016 2312 Doc.1521/4).]

  [L.V received funding for the research through the 'Soy in 1000 Garden Project'. B.V.B. was employed through this funding. The ‘Soy in 1000 Gardens’ project (GC03-C02) was funded by the Grand Challenges Program of VIB, which received support from the Flemish Government under the Management Agreement 2017-2021 (VR 2016 2312 Doc.1521/4).

- https://vib.be/en

- The sponsors played no role in the study design, data collection and analysis, decision to publish or preparation of the manuscript.]

4. We note that Figure 1 and 2 in your submission contain copyrighted images. All PLOS content is published under the Creative Commons Attribution License (CC BY 4.0), which means that the manuscript, images, and Supporting Information files will be freely available online, and any third party is permitted to access, download, copy, distribute, and use these materials in any way, even commercially, with proper attribution. For more information, see our copyright guidelines: http://journals.plos.org/plosone/s/licenses-and-copyright.

1. You may seek permission from the original copyright holder of Figure 1 and 2 to publish the content specifically under the CC BY 4.0 license. 

5. Please provide a complete Data Availability Statement in the submission form, ensuring you include all necessary access information or a reason for why you are unable to make your data freely accessible. If your research concerns only data provided within your submission, please write "All data are in the manuscript and/or supporting information files" as your Data Availability Statement.

Additional Editor Comments:

Dear Dr Bergen

Many thanks for submitting your MS to PLOS One I have now received 2 reviews of your paper,

This is an interesting paper which deals with an important issue: namely the determinants sustained of citizen engagement in citizen science activities in relation to agronomy. However, reviewer 1 has s have indicated that there issues in relation to methodological clarity which need to be addressed, recommending major revision. If you are able to address these comments, and those of reviewer 2, please submit a revised MS and cover letter detailing changes, revisions and authors' responses for further consideration for publication in this journal

Reviewers' comments:

Reviewer's Responses to Questions

**Comments to the Author**

1. Is the manuscript technically sound, and do the data support the conclusions?

Reviewer #1: Partly

Reviewer #2: Yes

2. Has the statistical analysis been performed appropriately and rigorously? 

Reviewer #1: I Don't Know

Reviewer #2: Yes

3. Have the authors made all data underlying the findings in their manuscript fully available?

Reviewer #1: No

Reviewer #2: Yes

4. Is the manuscript presented in an intelligible fashion and written in standard English?

Reviewer #1: Yes

Reviewer #2: Yes

5. Review Comments to the Author

Reviewer #1: Thank you for sharing the manuscript “What defines a citizen scientist and influences their sustained participation in an agronomic project?” with me. It presents original research on the personal variables that influence initial and sustained participation in an agricultural citizen science project. While the research design seems quite rigorous to me, the manuscripts’ claims how it extends previous research need to be substantiated. I have identified some gaps that concern the theoretical framing of the study and the robustness of the selection mechanism claimed (i.e., unidentified factors on behalf of the selecting project managers).

Introduction and Significance

In my view, how this study contribute to theory development is not yet clear for the following reasons, even though studies on the influence of motivation on initial and ongoing participation in citizen science are still relevant.

(1) The contribution of the study would be much clearer if you summarized the state of research on all constructs’ influences on participation, i.e. of knowledge, intention, concerns, values (and not only on motivation). What is the exact research gap related to motivation, knowledge, etc. in the context of initial and ongoing participation? Just stating that “how this [i.e. participation decision] relates to their personal characteristics […] is not entirely clear” (lines 63ff.) seems quite vague to me.

(2) The contribution to previous research would also be clearer if all personal characteristics could be located within a common theoretical framework. For this purpose, the representation of person characteristics in Figure 3 would have to be derived from the literature. Why are values and motivation equated here (in contrast to West and colleagues, 2021, for example)? Motivation can be based on values, but why should a value directly motivate me to act? Up to now, the personal characteristics seem to be a loose bundle that can not be subsumed under the umbrella term “motivations”.

In addition, the so-called project pipeline could be presented in a more differentiated way with reference to the relevant literature, from recruitment to initial participation to ongoing participation. Questions could be asked about who from the “general population” was not reached, who was reached but did not participate, who participated but did not follow through to the end, etc.

Methods

A possibly central problem with the procedure is that the selection model is supposed to predict self-selection on the basis of personal characteristics, although the selection was also made by the project managers (see lines 121f., 300). Could you substantiate that the selection models are nevertheless suitable?

The strengths of the constructs surveyed are that the individual scales are theory-based and have (largely) already been used in previous research. However, in my view, there is a lack of details on how the items were aggregated into scales: How did you, for example, aggregate the knowledge scales and why did you not calculate a CFA for the knowledge scales? How did you ensure that the items only load on the respective subscale and not on other subscales (if a one-factor model CFA is calculated for each subscale)? From my point of view, it is theoretically unfounded that all the scales surveyed are referred to as motivations (knowledge is not a motivation).

I cannot judge whether you adequately implemeted the Heckman model. However, it seems to me that the use of motivation as a variable that “affects why individuals may select to participate in study but does not influence the outcome variable” requires additional justifications (lines 352f.). First, why do you think that a t-test is an adequate measure for choosing the VFI variable and when do you consider the influence of a variable as small enough to not influence the outcome variable? Second, why did you differentiate between “end” and “partly” participants for the t-test while the selection models account for “Entry completion”, “Final step”, and “End”? I would either expect further t-tests to exclude influences on these groups, or only the two-step model with the “End” variable to be tested. Third, the well-documented influence of motivation in previous research speaks against the selection of the VFI variable (e.g., West et al., 2021), even if no differences between the groups were found in this study.

Please, make sure to share your data, for example, with your article in the repository of PLOS ONE.

Results and Claims

From a more formal point of view, the results are lacking important information. Please consider whether you would like to follow any reporting guideline (such as CROSS; https://www.ncbi.nlm.nih.gov/pmc/articles/PMC8481359/bin/11606_2021_6737_MOESM1_ESM.docx). Following such guidelines, you should include information on the estimates in the tables (means, standard deviations in Tables 4 and 5, coefficients in Table 6, etc.). For difference tests, the results should include an effect size (and confidence intervals). For the selection models, the results should include information on the model fit.

You state that the models correct for “non-randomly/self-selected” sample (lines 410f.). Could you include some estimate of the correction, especially in the light of missing information about influencing variables other than the included ones? That might be informative for other research to underline the importance of corrections. Furthermore, in the introduction, the selection mechanism should be clarified with regard to the literature (e.g., Nibble and drop framework; Fischer et al., 2020). Then, please consistently refer to either non-random or self-selection throughout the manuscript.

In my opinion, the discussion of the results falls short of the potential. This is also because the results are not discussed in sufficient detail against the background of the literature. The manuscript by Füchslin and colleagues (2019) offers a starting point for identifying further studies on selection factors. For example, with regard to knowledge as a variable, you could discuss whether, as in many other studies, the level of education also comes into play here (as the level of education and knowledge correlate at least to a certain extent).

Especially, for the two-step selection models, you should consider to further substantiate how your results extend previous research. Which influences from sustainable participation described in the literature persist or even disappear if selection is taken into account? Are there also influences that are reversed?

You might also want to discuss how reliable the results are if they are influenced by selection factors that lie outside the participant. Are there selection factors on behalf of the projects managers, either explicit to achieve a balanced sample or implicit biases? Furthermore, I could not find a Limitations section that could emphasize the strengths of the research approach but should also highlight the weaknesses.

Style and Language

The manuscript is well written and in fluent English, if one disregards some minor typos in the text and the tables (e.g., missing asterisk in Table 6). Please, check carefully the text again. Furthermore, it is of utmost importance that the manuscript coherently designates the constructs and variables examined in order to increase comprehensibility.

The title does not yet capture the contribution of the study, from my point of view: The manuscript does not define what a citizen science is, but what citizens’ characteristics sets sustained participation apart from initial participation.

Table 6 is not reader friendly because of the line breaks. Each variable should be in a single line.

The reporting of decimals should be reconsidered according to guidelines such as by the American Psychological Association (7th Edition): “… means and standard deviations for data measured on integer scales (e.g., surveys and questionnaires) to one decimal”; “other means and standard deviations and correlations, proportions, and inferential statistics (t, F, chi-square) to two decimals.” (https://apastyle.apa.org/instructional-aids/numbers-statistics-guide.pdf)

Literature

Fischer, H., Cho, H., & Storksdieck, M. (2021). Going Beyond Hooked Participants: The Nibble and- Drop Framework for Classifying Citizen Science Participation. Citizen Science: Theory and Practice, 6(1), 10. https://doi.org/10.5334/cstp.350

Füchslin, T., Schafer, M. S., & Metag, J. (2019). Who wants to be a citizen scientist? Identifying the potential of citizen science and target segments in Switzerland. Public Understanding of Science, 28(6), 652-668. https://doi.org/10.1177/0963662519852020

West, S., Dyke, A., & Pateman, R. (2021). Variations in the Motivations of Environmental Citizen Scientists. Citizen Science: Theory and Practice, 6(1), Article 14. https://doi.org/10.5334/cstp.370

Reviewer #2: This manuscript presents a very comprehensible study with an important research question, a clear and clever research design, and interesting and convincing results. Accordingly, I don’t have any comments worth mentioning about the study itself.

What I’m not really satisfied with yet, on the other hand, are the introductory reflections on the role of people’s motivations and reasons to engage in Citizen Science – and their motivation then actually to maintain their commitment to a CS project. There is ample and recent research that the authors should check out for improving their introduction and that they should compare their own findings to in their discussion section:

Greving and colleagues, for example, have identified several relevant psychological factors of influence for CS participants. There is research on the influence of psychological ownership on pride in a Citizen Science project on wildlife ecology. There are also findings that show that CS participants attitudes toward engagement in Citizen Science increase self-related, ecology-related, and motivation-related outcomes in an urban wildlife project.

Moreover, CS participation had an impact on attitudes and knowledge in a Citizen Science project about urban bat ecology. Finally, in lab studies it was also found that people’s compassion is relevant for CS participation.

Considering these previous findings and positioning their own research more thoroughly in the existing literature would benefit the authors’ presentation of their study. This could strengthen their line of argument and, in particular, could help them better justify their selection of relevant variables.

6. PLOS authors have the option to publish the peer review history of their article (what does this mean?). If published, this will include your full peer review and any attached files.

Reviewer #1: **Yes: **Till Bruckermann

Reviewer #2: No

---

## [Author Response · Author response to Decision Letter 0]

12 Apr 2024

A general remark from the authors: to be able to include Table 6 in Landscape orientation in the Word document, the paper is divided into 2 “Word-sections”. As a result, after Table 6, Line-numbers start again from 1. This is something to keep in mind when reading the line-references in the following paragraphs of the rebuttal.

General PlosOne style/formatting/etc comments:

Please ensure that your manuscript meets PLOS ONE's style requirements, including those for file naming. The PLOS ONE style templates can be found at https://journals.plos.org/plosone/s/file?id=wjVg/PLOSOne_formatting_sample_main_body.pdf and https://journals.plos.org/plosone/s/file?id=ba62/PLOSOne_formatting_sample_title_authors_affiliations.pdf. 

We have adapted the file names of figures and tables according to PlosOne guidelines.

2. Please provide additional details regarding participant consent. 

In the ethics statement in the Methods and online submission information, please ensure that you have specified (1) whether consent was informed and (2) what type you obtained (for instance, written or verbal, and if verbal, how it was documented and witnessed). If your study included minors, state whether you obtained consent from parents or guardians. If the need for consent was waived by the ethics committee, please include this information.

We have added that written, online informed consent was obtained from each participant in the study on line 304.

[The authors thank all volunteers, including those candidates who were not admitted to the project, for their enthusiasm to take part in the citizen science project and the collection of the survey and garden data. We also thank all partners and co-researchers on the project: VIB-UGent Center for Plant Systems Biology (Prof. Sofie Goormachtig, Prof. Steven Maere, Dr. Stien Mertens, Dr. Lena Vlaminck, Dr. Sonia Garcia Mendez, Alexander Clarysse), ILVO (Prof. Isabel Roldán-Ruiz, Dr. Joke Pannecoucque, Margo Vermeersch, Koen Van Loo), Ghent University (Prof. Anne Willems, Helena Van den Eynde, Ilse de Baenst, Prof. Caroline De Tender), VIB-KU Leuven (Prof. Jan Michiels, Serge Beullens), VIB HQ (Dr. Astrid Gadeyne, Dr. Laurien Van Dyck, Merlijn Vanhecke, Teknorix), MijnTuinlab, Bodemkundige Dienst van België. The ‘Soy in 1000 Gardens’ project (GC03-C02) has received funding from the Grand Challenges Program of VIB, which received support from the Flemish Government under the Management Agreement 2017-2021 (VR 2016 2312 Doc.1521/4).]

We note that you have provided funding information that is not currently declared in your Funding Statement. However, funding information should not appear in the Acknowledgments section or other areas of your manuscript. We will only publish funding information present in the Funding Statement section of the online submission form. Please remove any funding-related text from the manuscript and let us know how you would like to update your Funding Statement. Currently, your Funding Statement reads as follows: [L.V received funding for the research through the 'Soy in 1000 Garden Project'. B.V.B. was employed through this funding. The ‘Soy in 1000 Gardens’ project (GC03-C02) was funded by the Grand Challenges Program of VIB, which received support from the Flemish Government under the Management Agreement 2017-2021 (VR 2016 2312 Doc.1521/4).

- https://vib.be/en

- The sponsors played no role in the study design, data collection and analysis, decision to publish or preparation of the manuscript.]

We have removed the funding information from the acknowledgements. We do not wish to make any updates to the Funding Statement section of the online submission form as all funding information is included.

4. We note that Figure 1 and 2 in your submission contain copyrighted images. 

All PLOS content is published under the Creative Commons Attribution License (CC BY 4.0), which means that the manuscript, images, and Supporting Information files will be freely available online, and any third party is permitted to access, download, copy, distribute, and use these materials in any way, even commercially, with proper attribution. For more information, see our copyright guidelines: http://journals.plos.org/plosone/s/licenses-and-copyright.

You may seek permission from the original copyright holder of Figure 1 and 2 to publish the content specifically under the CC BY 4.0 license. We recommend that you contact the original copyright holder with the Content Permission Form (http://journals.plos.org/plosone/s/file?id=7c09/content-permission-form.pdf) and the following text:

 l“I request permission for the open-access journal PLOS ONE to publish XXX under the Creative Commons Attribution License (CCAL) CC BY 4.0 (http://creativecommons.org/licenses/by/4.0/). Please be aware that this license allows unrestricted use and distribution, even commercially, by third parties. Please reply and provide explicit written permission to publish XXX under a CC BY license and complete the attached form.”

Figure 1 and 2 have been removed from the submission, and instead one new, original Figure 1 has been added.

5. Please provide a complete Data Availability Statement in the submission form, ensuring you include all necessary access information or a reason for why you are unable to make your data freely accessible. If your research concerns only data provided within your submission, please write "All data are in the manuscript and/or supporting information files" as your Data Availability Statement.

The dataset used for this publication is currently submitted and under review at the Research Data Repository (RDR) by KU Leuven. Once reviewed and approved the data will be published under the following DOI: https://doi.org/10.48804/UYEYVY . As the data is still under review, currently, this DOI is not yet active.

The full name of the ethics committee which approved the study, the Social and Societal Ethics Committee (SMEC) of KU Leuven is present in the paper on line 303. We have added that written, online informed consent was obtained from each participant in the study on line 304.

Captions for the Supporting Information files have been added at the end of the manuscript on lines 451-464 of Section 2 of the paper, which starts after Table 6. The paper was divided into two sections, in order to fit Table 6 in Landscape orientation between both sections.

Reviewer #1: Thank you for sharing the manuscript “What defines a citizen scientist and influences their sustained participation in an agronomic project?” with me. It presents original research on the personal variables that influence initial and sustained participation in an agricultural citizen science project. While the research design seems quite rigorous to me, the manuscripts’ claims how it extends previous research need to be substantiated. I have identified some gaps that concern the theoretical framing of the study and the robustness of the selection mechanism claimed (i.e., unidentified factors on behalf of the selecting project managers).

1. Introduction and Significance

In my view, how this study contribute to theory development is not yet clear for the following reasons, even though studies on the influence of motivation on initial and ongoing participation in citizen science are still relevant.

(A) The contribution of the study would be much clearer if you summarized the state of research on all constructs’ influences on participation, i.e. of knowledge, intention, concerns, values (and not only on motivation). What is the exact research gap related to motivation, knowledge, etc. in the context of initial and ongoing participation? Just stating that “how this [i.e. participation decision] relates to their personal characteristics […] is not entirely clear” (lines 63ff.) seems quite vague to me.

We thank the reviewer for these suggestions and agree that we should adapt the introduction to more clearly indicate the state-of-research. We completely reworked the introduction, and added section titles to concretely discuss, among others, the quantification of participation in citizen science, as well as factors influencing initial and sustained participation (Line 68-159). Further, we also expanded on the current research gap and introduced the use of the Heckman selection model more extensively in introduction (Line 166-198). Because of the changes and additions, the introduction has become quite lengthy, but we believe all the information to be important for a comprehensive oversight of the state-of-the-art.

(B) The contribution to previous research would also be clearer if all personal characteristics could be located within a common theoretical framework. For this purpose, the representation of person characteristics in Figure 3 would have to be derived from the literature. Why are values and motivation equated here (in contrast to West and colleagues, 2021, for example)? Motivation can be based on values, but why should a value directly motivate me to act? Up to now, the personal characteristics seem to be a loose bundle that cannot be subsumed under the umbrella term “motivations”.

We agree with the reviewer that using the term motivations as an umbrella-term was not the correct way to refer to all the influencing variables. As such, we have opted for splitting the factors in two categories, namely ‘motivations’ and ‘dispositional variables’ after West et al., 2016 and Penner 2002. Dispositional variables encompass “all attributes of individuals influencing their likelihood of volunteering, including personality traits, personal beliefs, values, and demographic characteristics such as age, income, and education”. The term is introduced in the literature study (Line 178-185). From thereon out we consistently use this term throughout the rest of the paper. We also adapted Fig. 3, which has now become Fig. 2, accordingly.

(C) In addition, the so-called project pipeline could be presented in a more differentiated way with reference to the relevant literature, from recruitment to initial participation to ongoing participation. 

We thank the reviewer for this suggestion and agree that it is more relevant to present the figure in way that reflects the different stages of recruitment and retention. We adapted the figure completely, partly inspired by the ‘Nibble-and-Drop’-framework by Fischer (Fig. 1).

(D) Questions could be asked about who from the “general population” was not reached, who was reached but did not participate, who participated but did not follow through to the end, etc.

We agree with the reviewer that those are relevant questions to ask, and have added a section on this topic in the Limitations section of the Discussion (Line 157-169).

2. Methods

A) A possibly central problem with the procedure is that the selection model is supposed to predict self-selection on the basis of personal characteristics, although the selection was also made by the project managers (see lines 121f., 300). Could you substantiate that the selection models are nevertheless suitable?

We apologize for not making ourselves entirely clear. The term self-selection refers to the act of self-selection of the participants into the pool of citizen scientists by putting themselves up as candidates for the project. Previous research (see references at the end of the paragraph) has already used two-step models in both volunteering and citizen science for this exact “problem”. As a result, and because a general population sample was at hands, we opted for using a two-step method in order to correct for this potential self-selection bias. From this self-selected pool of citizen scientists, the project management then selected a balanced sample based on age, gender and educational level. 

We added the explanation of participant recruitment and selection by managers into Materials and Methods at line 264-280. We also added a more in depth explanation on the Heckman and our choice for this technique from line 462-468.

References:

Cameron TA, Kolstoe SH. Using Auxiliary Population Samples for Sample-Selection Correction in Models Based on Crowd-Sourced Volunteered Geographic Information. Land Econ. 2022;98(1):1–21. 

Handy F, Cnaan RA, Hustinx L, Kang C, Brudney JL, Haski-Leventhal D, et al. A Cross-Cultural Examination of Student Volunteering: Is It All About Résumé Building? Nonprofit Volunt Sect Q. 2010;39(3):498–523. 

B) The strengths of the constructs surveyed are that the individual scales are theory-based and have (largely) already been used in previous research. However, in my view, there is a lack of details on how the items were aggregated into scales: How did you, for example, aggregate the knowledge scales and why did you not calculate a CFA for the knowledge scales? How did you ensure that the items only load on the respective subscale and not on other subscales (if a one-factor model CFA is calculated for each subscale)? From my point of view, it is theoretically unfounded that all the scales surveyed are referred to as motivations (knowledge is not a motivation).

We apologize for not being entirely clear, and have adapted the text to explicitly explain that the scales were aggregated using the average of the items used per latent factor (Lines 340-341, 347-348, 358-359, 406, and 412). As mentioned by the reviewer all used scales have been widely used, and tested and validated extensively. This is why we opted for performing CFA’s to confirm pre-defined fits. Multiple factor CFA within one scale (e.g. VFI as a whole instead of looking at Values only) showed similar results, and gave no additional about item loadings on other latent factors besides the specified one. Additionally, doing an EFA to look at loadings outside of item’s resp

---

## [Editor Report · Decision Letter 1]

19 Apr 2024

Citizen scientists: unveiling motivations and characteristics influencing initial and sustained participation in an agricultural project

PONE-D-23-34026R1

Dear Dr. Van den Berghen

We’re pleased to inform you that your manuscript has been judged scientifically suitable for publication and will be formally accepted for publication once it meets all outstanding technical requirements.

Kind regards,

Lynn Jayne Frewer, MSc PhD

Academic Editor

PLOS ONE

---

## [Editor Report · Acceptance letter]

9 May 2024

PONE-D-23-34026R1 

PLOS ONE

Dear Dr. Vanden Berghen, 

I'm pleased to inform you that your manuscript has been deemed suitable for publication in PLOS ONE. Congratulations! Your manuscript is now being handed over to our production team.

Kind regards, 

on behalf of

Dr. Lynn Jayne Frewer 

Academic Editor

PLOS ONE